# Burden of mental health problems among pregnant and postpartum women in sub-Saharan Africa: systematic review and meta-analysis protocol

Elizabeth Awini,[1] Irene Akua Agyepong,[1,2] David Owiredu,[3] Leveana Gyimah,[4,5] Mary Eyram Ashinyo,[6] Linda Lucy Yevoo,[1] Sorre Grace Emmanuelle Victoire Aye,[1,2] Shazra Abbas,[7] Anna Cronin de Chavez  ,[8] Sumit Kane  ,[7] Tolib Mirzoev  ,[8] Anthony Danso-Appiah  [3,9]

**Correspondence to**
Anthony Danso-Appiah;
adanso-appiah@ug.edu.gh

## ABSTRACT

**Introduction** Pregnancy and postpartum-related mental health problems pose serious public health threat to the society, but worryingly, neglected in sub-Saharan Africa (SSA). This review will assess the burden and distribution of maternal mental health (MMH) problems in SSA, with the aim to inform the implementation of context sensitive interventions and policies.

**Methods and analysis** All relevant databases, grey literature and non-database sources will be searched. PubMed, LILAC, CINAHL, SCOPUS and PsycINFO, Google Scholar, African Index Medicus, HINARI, *African Journals Online* and IMSEAR will be searched from inception to 31 May 2023, without language restriction. The reference lists of articles will be reviewed, and experts contacted for additional studies missed by our searches. Study selection, data extraction and risk of bias assessment will be done independently by at least two reviewers and any discrepancies will be resolved through discussion between the reviewers. Binary outcomes (prevalence and incidence) of MMH problems will be assessed using pooled proportions, OR or risk ratio and mean difference for continuous outcomes; all will be presented with their 95% CIs. Heterogeneity will be investigated graphically for overlapping CIs and statistically using the $I^2$ statistic and where necessary subgroup analyses will be performed. Random-effects model meta-analysis will be conducted when heterogeneity is appreciable, otherwise fixed-effect model will be used. The overall level of evidence will be assessed using Grading of Recommendations Assessment, Development and Evaluation.

**Ethics and dissemination** Although no ethical clearance or exemption is needed for a systematic review, this review is part of a larger study on maternal mental health which has received ethical clearance from the Ethics Review Committee of the Ghana Health Service (GHS-ERC 012/03/20). Findings of this study will be disseminated through stakeholder forums, conferences and peer review publications.

**PROSPERO registration number** CRD42021269528.

## STRENGTHS AND LIMITATIONS OF THIS STUDY

⇒ This study uses robust methods, best practices and reporting guidelines to attempt to synthesise evidence on the burden of maternal mental health problems in sub-Saharan Africa to provide country and regional-specific estimates.

⇒ Screening of articles, data extraction and quality assessment will be done using validated tools by at least two independent reviewers to minimise bias.

⇒ The study uses comprehensive search terms and strategy, and involves relevant electronic databases and non-database sources to attempt to retrieve all potentially relevant studies.

⇒ A possible limitation is, there were no previous data to compare our systematic review to.

## INTRODUCTION

Maternal mental health (MMH) problems, occurring during pregnancy and postpartum constitute a serious public health problem globally, affecting about 10% of pregnant women and 13% of postpartum mothers.[1–3] A systematic review on global depression among postpartum women estimated prevalence of 17.2% worldwide.[4] In high-income countries, prevalence of postpartum depression ranged from 5% to 20%.[5] Data from 2010 to 2015 Northern Ireland Maternity System indicated that 18.9% of pregnant women reported a history of at least one mental disorder.[6] In the USA, 10% depression in mothers was reported over 12 months.[7] Analysis of Claims data from January to December 2008 involving 38 174 pregnant women in Germany identified at least one mental health problem from four main mental health disorders in 16 639 of the women[8] with somatoform/dissociative

BMJ

disorder being 24.2%, anxiety 16.9%, stress reactions 11.7% and depression 9.3% . While reliable data are not readily available, existing estimates suggest that the burden of MMH problems is relatively higher in low-income and middle-income countries (LMICs) with one in four women reporting depression during pregnancy and 1 in five after delivery.[9] A review of evidence from LMICs on prenatal and postnatal depression reported prevalence of 4.9%–50%,[10] whereas a systematic review involving prenatal and postnatal women living in Africa reported prevalence of depression during pregnancy at 11.3% and 18.3% after delivery.[11] Another review in sub-Saharan Africa (SSA) reported 18.6% postpartum depression,[12] with a range from 7% to 50.3%.

The common mental health problems experienced by pregnant and postpartum women are depression and anxiety.[13–16] Depression can be mild, moderate or severe. Mild depression will normally not affect the individual's ability to undertake their day-to-day activities such as self-care and interpersonal relationships, whereas moderate to severe episodes can render the mother less capable of undertaking their basic self-care and that of their newborn babies which could impact on breastfeeding and bonding.[17] Other MMH disorders are postpartum psychosis, pregnancy and postpartum obsessive–compulsive disorder (OCD), birth-related post-traumatic stress disorder (PTSD), intrusive thoughts and mania, schizophrenia, infanticide, substance use disorder, anorexia nervosa, bulimia, suicidal ideation, bipolar affective disorder, paranoia, psychopathy, neurotic disorders and self-harm.[18 19] These disorders may develop as a result of experiences associated with childbirth, foetal loss, congenital malformations, intimate partner violence during pregnancy, lack of partner support, history of abuse, unplanned pregnancy, complications in previous pregnancy, higher perceived stress, lower self-esteem[20–24] and many more. Individuals with predisposition to bipolar affective disorders may develop manic episodes from pregnancy-related stress.[20] Severity of MMH disorders can potentially progresses from mild or moderate to severe forms if not identified early and appropriate intervention instituted.[25 26]

### Rationale for this systematic review
Mothers with mental health problems may not be able to realise their abilities, work productively or cope with the normal stresses of life.[3] MMH disorders can have negative effects on both the mother and the child.[27] In severe cases, depressed mothers may have symptoms of a psychosis.[4] Affected mothers often cannot function properly, with some having suicidal tendencies.[3] Evidence shows that MMH problems are associated with negative birth outcomes and may adversely affect cognitive development of the infant,[28 29] nutritional status of their infants[30] and early child well-being.[31] There is, however, limited knowledge to inform policy makers on the burden of MMH disorder in SSA to inform on the selection and implementation of locally relevant, feasible and effective

interventions. MMH is an important health problem which should be given the needed attention but this has often been neglected as a health priority.[32–34] There is much evidence that in LMICs, the vast majority of women who experience mental health problems during and after pregnancy do not receive the needed treatment.[1 35] Although few systematic reviews have investigated mental health problems in SSA, none focused specifically on MMH.[36 37] The only review that assessed risk factors for antenatal depression, included only studies conducted in Ethiopia,[36] and although a systematic review protocol has been registered in PROSPERO,[37] it intends to explore the role of MMH disorders and stillbirths.

This review will assess the burden of MMH problems among pregnant and postpartum women in SSA and attempts to provide robust country and regional estimates of the burden of disease across countries in SSA. Specifically, it will determine the prevalence and incidence and describe the sociodemographic characteristics, general obstetric histories and characteristics of women with MMH problems; assess effects of MMH problems on birth outcomes and analyse for differences in the burden of mental health problems among pregnant and postpartum women between rural and urban settings. The review will answer the following specific questions: (1) What is the magnitude of MMH problems among pregnant and postpartum women living in SSA? (2) What are the sociodemographic characteristics of pregnant and postpartum women with mental health problems in SSA? (3) What are the general obstetric histories and characteristics of pregnant and postpartum women in SSA with mental health problems? (4) Is there any difference in the burden of pregnant and postpartum women with mental health problems between rural and urban settings? and (5) Are there any documented effects of birth outcomes of pregnant women with mental health problems?

### Review methods
This review protocol has been prepared following the Preferred Reporting Items for Systematic Review and Meta-Analysis extension for protocols (PRISMA-P)[38] (online supplemental file 1) and the PRISMA flow diagram (online supplemental file 2). The full review will be prepared in line with the Preferred Reporting Items for Systematic Review and Meta-Analysis (PRISMA). The full review is expected to start on 1 May 2023, analysis 1 September and completed by 30 November 2023.

### Patient and public involvement
The review questions and outcome measures have been developed collaboratively with the relevant patient and consumer involvement and informed by their priorities, experience and preferences in line with GRIPP2 reporting checklists. The review findings will be shared with relevant wider patient communities who will also be involved in the results dissemination.

## Criteria for considering studies for this review

### Types of studies

Any study (cohort, case–control and cross-sectional studies) conducted in SSA that assessed burden of MMH problems among pregnant and postpartum women will be eligible for inclusion. This review is not an intervention effectiveness review and randomised controlled trials (RCTs) are not the focus but if a RCT reported baseline number of cases and used a well-defined sample to serve as the denominator to allow the calculation of proportion/prevalence/incidence in pregnant or postpartum women, such an RCT will be eligible for inclusion. Reviews will not be eligible for inclusion. However, we will go through the reviews to identify potentially eligible studies missed by our searches. If the study is a global review having, for example, SSA or subregional subset, we will extract the studies conducted in SSA for inclusion in this review. If the study reported a country or regional estimate without a well-defined sample (representative sample or subsample of the source population), it will not be eligible for inclusion. In cases where the results of a multicountry study have been lumped together and there is no way of disaggregating the data, such studies will not be included. Case studies and case series (these are atypical and not representative of the source population), commentaries or opinions, will not be eligible for inclusion.

### Participants

Pregnant and postpartum (postpartum is defined as up to 12 months after delivery) women living in SSA, diagnosed of any mental health disorder (depression, anxiety, postpartum psychosis, bipolar disorders, substance misuse disorders, dysthymia, OCD, PTSD, schizophrenia, infanticide, suicidal ideation, paranoia, psychopathy, neurosis/neurotic disorders, self-harm, anorexia nervosa, bulimia, etc.) will be eligible for inclusion. The tool or criteria for diagnosis should be stated, for example, standard operational diagnostic criteria such as Research Diagnostic Criteria (RDC),[39] the 10th edition of the International Classification of Diseases (ICD-10)[40] or Diagnostic and Statistical Manual of Mental Disorders (DSM-V).[41] Pregnant or postpartum women whose diagnosis of mental health disorder could not be confirmed will be excluded.

### Interventions

This systematic review is not intervention review.

### Comparison

This is non-comparative review but where outcomes or variables permit comparison, we will attempt to compare.

### Outcomes

#### Primary outcomes

► Burden of MMH problems measured as prevalence, incidence, etc.
► Birth outcomes (maternal and foetal outcomes) up to 12 months postpartum
► Type of MMH conditions (for diagnosed cases)

#### Secondary outcomes

► Proportion of MMH conditions captured in the community
► Predisposition socio-demographic characteristics of pregnant and postpartum women with mental health problems
► Predisposition obstetric histories and characteristics of pregnant and postpartum women in SSA with mental health problems
► Psychiatric predisposition (pre-existing mental health issues triggered during or after pregnancy).
► Burden of mental health problems among pregnant and post-partum women between rural and urban settings

### Search strategy

We will search the following electronic databases: PubMed, PsycINFO, LILAC and CINAHL from inception to 31 May 2023 without language restriction and using the search terms in table 1. We will also search Google Scholar, African Index Medicus, HINARI, *African Journals Online*, IMSEAR and relevant preprint repositories. Grey literature including dissertations and conference proceedings will be searched. Reference list of retrieved articles will be reviewed, and experts in the field of MMH will be contacted for studies not captured by our searches (see table 1 for search strategy developed for PubMed).

The search strategy will be adapted as appropriate for other databases. All searches will be rerun just before the final analyses and any further eligible studies identified will be included.

### Study selection

The search output will be managed, collated and deduplicated using EndNote. The deduplicated articles will be exported to Rayyan[42] for screening and selection. At least two reviewers will screen titles and abstracts of identified studies independently using prespecified and piloted eligibility flowchart (figure 1). The full text of potentially relevant studies will also be reviewed independently for inclusion. The PRISMA flow diagram will be used to document the flow of studies and reasons for exclusion. Discrepancies will be resolved through discussion between the reviewers.

### Data extraction and management

A validated data extraction sheet adapted from the Cochrane Collaboration (online supplemental file 3) will be used by the reviewers to independently extract relevant data. Data to be extracted include characteristics of the studies such as year study was conducted, year study was published (for published studies), country where study was conducted, study design and sample size; sociodemographic characteristics of the participants (age, setting, socioeconomic status, level of education and occupation); obstetric factors (parity, maternal age, age at first delivery, history of miscarriage and history of stillbirth), mental health conditions (depression, anxiety,

| Table 1 | Search strategy for PubMed (to be adapted for the other databases) | |
|---|---|---|
| Search | Query | Results |
| #1 | Search: ((((((((((((((((((((((('mental health'(Title/Abstract)) OR ('mental disorder'(Title/Abstract))) OR ('mental illness'(Title/Abstract))) OR ('mental problem'(Title/Abstract))) OR ('chronic mental illness'(Title/Abstract))) OR ('psychiatric illness'(Title/Abstract))) OR ('chronic psychiatric illness'(Title/Abstract))) OR ('chronic insanity'(Title/Abstract))) OR (insanity(Title/Abstract))) OR ('chronic mental disorder'(Title/Abstract))) OR ('dementia praecox'(Title/Abstract))) OR (schizophrenia(Title/Abstract))) OR (psychoses(Title/Abstract))) OR (psychosis(Title/Abstract))) OR ('schizophrenic psychosis'(Title/Abstract))) OR (depression(Title/Abstract))) OR ('mental sickness'(Title/Abstract))) OR ('mental disease'(Title/Abstract))) OR (maladjustment(Title/Abstract))) OR ('emotional disorder'(Title/Abstract))) OR ('nervous disorder'(Title/Abstract))) OR ('nervous breakdown'(Title/Abstract))) OR (neurosis(Title/Abstract))) OR ('neurotic disorder'(Title/Abstract))) OR (psychopathy(Title/Abstract))) OR (paranoia(Title/Abstract)) | |
| #2 | Search: (((((((((((('pregnant women'(Title/Abstract)) OR (prepartum(Title/Abstract))) OR (peripartum(Title/Abstract))) OR (perinatal(Title/Abstract))) OR (prenatal(Title/Abstract))) OR (antenatal(Title/Abstract))) OR ('during pregnancy'(Title/Abstract))) OR (postpartum(Title/Abstract))) OR (maternal(Title/Abstract))) OR (maternity(Title/Abstract))) OR (parturient(Title/Abstract))) OR (antepartum(Title/Abstract))) OR ('post-delivery'(Title/Abstract))) OR (puerperium(Title/Abstract)) | |
| #3 | Search: (#1) AND (#2) | |
| #4 | Search: (((((((((((((((((((((((((((((((((((((('sub-Saharan Africa') OR (SSA)) OR (Angola)) OR (Benin)) OR (Botswana)) OR ('Burkina Faso')) OR (Burundi)) OR (Cameroon)) OR ('Cape Verde')) OR ('Central African Republic')) OR (Chad)) OR (Comoros)) OR (Congo)) OR ('Cote d'Ivoire')) OR (Djibouti)) OR ('Equatorial Guinea')) OR (Ethiopia)) OR (Gabon)) OR ('The Gambia')) OR (Ghana)) OR (Guinea)) OR ('Guinea-Bissau')) OR (Kenya)) OR (Lesotho)) OR (Liberia)) OR (Madagascar)) OR (Malawi)) OR (Mali)) OR (Mauritania)) OR (Mauritius)) OR (Mozambique)) OR (Namibia)) OR (Niger)) OR (Nigeria)) OR (Rwanda)) OR ('Sao Tome and Principe')) OR (Senegal)) OR (Seychelles)) OR ('Sierra Leone')) OR (Somalia)) OR ('South Africa')) OR (Sudan)) OR (Swaziland)) OR (Tanzania)) OR (Togo)) OR (Uganda)) OR (Zaire)) OR (Zambia)) OR (Zimbabwe) | |
| #5 | Search: (#3) AND (#4) | |

postpartum psychosis, dysthymia/persistent depressive disorder, pregnancy and postpartum OCD, birth-related PTSD, intrusive thoughts and mania, schizophrenia, infanticide, substance use disorder, anorexia nervosa, bulimia, suicidal ideation, bipolar affective disorder, paranoia, psychopathy, neurotic disorders and self-harm) and quality domains for the assessment of quality of the included studies for risk of bias (ROB). Data on burden of the disease (such as prevalence, incidence and duration of the MMH problem) will be extracted. The corresponding authors of the primary studies will be contacted for missing data or unclear information. Where it is not possible to obtain the missing information, data will be analysed based on those with complete outcome data and the amount of data missing with reasons will be provided. If necessary, data will be coded and recoded before use in the analysis. The extracted data will be verified independently and any disagreement will be resolved through discussion.

## Assessment of quality of the included studies

At least two reviewers will assess quality of the included studies for ROB (methodology and reporting) independently using the appropriate ROB assessment tools. Since this review is not focusing on randomised controlled trials (RCTs), the domains on the Cochrane ROB tool[43] will not be covered fully. Selective outcome and analysis reporting bias (which occurs when studies with positive and significant results are likely to be reported or published) are the domains to be considered on the Cochrane ROB tool. The Cochrane criteria for reporting ROB in the included studies will be used to assess prespecified outcomes of interest. The ROB will be rated as 'low' if protocol of the study is available and all prespecified outcomes of interest as specified in the protocol, or the protocol is not available but it is clear that all prespecified and expected outcomes of interest have been reported. The ROB will be rated as 'high' for a study if outcomes are not reported as prespecified or expected and 'unclear' when there is not enough information to make clear judgement. Other prespecified biases pertaining to the methods, source of funding, etc, will be assessed and rated. The ROB tool for prevalence studies will be assessed using the tool by Hoy et al[44] (online supplemental file 4) on four domains: selection bias, non-response bias, measurement bias and bias related to data analysis for prevalence studies. Each ROB domain will be graded as 'low risk', 'high risk' and 'unclear' ROB. Any disagreements will be resolved through discussion between the reviewers, and if necessary, a third reviewer will be consulted.

## Data analysis

Binary outcomes will be assessed using OR or risk ratio (RR), and for continuous data, we will use mean difference (MD) or standardised mean difference (SMD) for means that used different scales. Meta-analysis, the statistical component of systematic review, will be used to combine study outcomes. OR, RR and MD of the individual studies will be pooled and presented with their

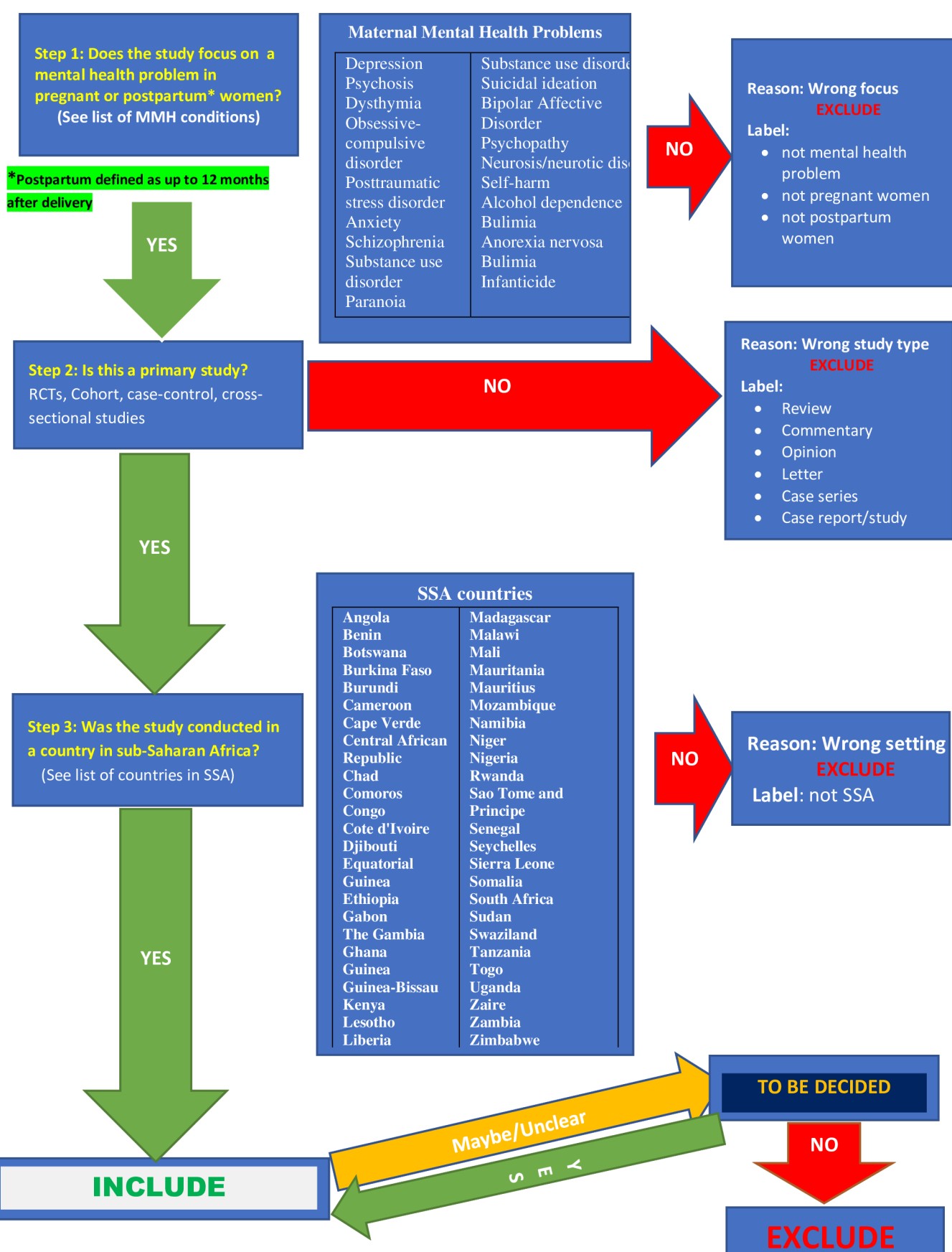

**Figure 1** Study selection flowchart.

95% CI. Random-effects model will be used in the meta-analysis when heterogeneity is high, otherwise fixed-effect model will be used. Descriptive statistics will be used to describe proportions (prevalence and incidence).

## Heterogeneity and subgroup analysis

Heterogeneity arises because of variation in the study design, characteristics of participants or outcomes between or within studies.[45] Heterogeneity will be investigated both graphically and statistically. The $I^2$ statistic which describes the percentage of variability that is due to heterogeneity rather than chance will be estimated. The $I^2$ is classified into four levels: 0%, 1%–29%, 30%–59% and 60%–100%[46] and $I^2 > 50\%$ indicates significant heterogeneity.[47] Subgroup analysis which is used to estimate an effect of indicator within each subgroup or subset will be used to address heterogeneity due to variation in effects in each subgroup mixed-effect model. Thus, the burden of MMH problems will be estimated separated for urban and rural and for each age group. Subgroup analysis will also be done for gestational age and postpartum period. If we find sufficient number of studies, subgroup analysis will also be based on perinatal mental disorder type, study design, parity and location.

## Grading the evidence

The overall evidence of the systematic review will be graded using Grading of Recommendations Assessment, Development and Evaluation (GRADE)[48] (available from guidelinedevelopment.org/handbook). The GRADE system assesses the following domains: ROB, imprecision, inconsistency, indirectness and publication bias, and classifies the quality of evidence as high, moderate, low and very low. If evidence from a study is graded high quality, it implies further research is very unlikely to change the confidence in the estimate of effect while a grading of very low quality implies an estimate of effect is very doubtful.

## ETHICAL APPROVAL AND DISSEMINATION

Although no ethical clearance or exemption is needed for a systematic review, this review is part of a larger study on MMH that involves primary data collection and which has received ethical clearance from the ethics review committee of the Ghana Health Service (GHS-ERC 012/03/20). The results of the review will be presented to stakeholders (policymakers and practitioners). It will also be disseminated through conferences and peer review publications.

**Author affiliations**
[1]Research and Development Division, Dodowa Health Research Centre, Ghana Health Service, Dodowa, Ghana
[2]Faculty of Public Health, Ghana College of Physicians and Surgeons, Accra, Ghana
[3]Department of Epidemiology and Disease Control, School of Public Health, University of Ghana, Legon, Ghana
[4]Department of Psychiatry, Pantang Hospital, Accra, Ghana
[5]Faculty of Psychiatry, Ghana College of Physicians and Surgeons, Accra, Ghana
[6]Institutional Care Division, Ghana Health Service, Accra, Ghana
[7]Nossal Institute for Global Health, Melbourne School of Population and Global Health, The University of Melbourne, Melbourne, Victoria 3010, Australia
[8]Department of Global Health and Development, London School of Hygiene and Tropical Medicine, London, UK
[9]Centre for Evidence Synthesis and Policy, University of Ghana, Legon, Accra, Ghana

**Acknowledgements** We thank Dr Caleb Othieno, University of Nairobi, Kenya, Mrs Ruth Owusu-Antwi, Komfo Anokye Teaching Hospital, Kwame Nkrumah University of Science and Technology, Kumasi, Ghana and Stephanie Catsaros, Université Paris Cité, France for peer-reviewing this manuscript and providing very useful comments. This systematic review was part of capacity building initiative by the UG Centre for Evidence Synthesis and Policy (UGCESP), Africa Communities of Evidence Synthesis and Translation (ACEST) and the RESPONSE Project that are jointly training experts in evidence synthesis and translation across low-income and middle-income countries (LMICs).

**Contributors** Protocol conceptualisation and design: EA, IAA, SK, TM and AD-A. Drafting the work or revising it critically for important intellectual content: EA, IAA, DO, LG, MEA, LY, SGEVA, SA, ACC, SK, TM and AD-A. Acquisition, analysis or interpretation of data: EA, IAA, DO, LG, MEA, LY, SGEVA, SA, ACC, SK, TM and AD-A. Agreement to be accountable for accuracy or integrity of all aspects of the work: EA, IAA, DO, LG, MEA, LY, SGEVA, SA, ACC, SK, TM and AD-A. Final approval of the version to be published: EA, IAA, DO, LG, MEA, LY, SGEVA, SA, ACC, SK, TM and AD-A. Supervised this work: AD-A.

**Funding** This review is part of a wider RESPONSE study, which received funding from the Joint MRC/ESRC/DFID/Wellcome Health Systems Research Initiative (grant ref: MR/T023481/2). The views expressed in this publication are those of the author(s) and not necessarily those of the funders.

**Competing interests** None declared.

**Patient and public involvement** Patients and/or the public were involved in the design, or conduct, or reporting or dissemination plans of this research. Refer to the Methods section for further details.

**Patient consent for publication** Not required.

**Provenance and peer review** Not commissioned; externally peer reviewed.

**Data availability statement** Data sharing not applicable as no data sets generated and/or analysed for this study. All data relevant to the study are included in the article or uploaded as supplementary information.

**ORCID iDs**
Anna Cronin de Chavez http://orcid.org/0000-0002-4050-4276
Sumit Kane http://orcid.org/0000-0002-4858-7344
Tolib Mirzoev http://orcid.org/0000-0003-2959-9187
Anthony Danso-Appiah http://orcid.org/0000-0003-1747-0060

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
