## [Reviewer comments · BMJ Open]

ARTICLE DETAILS

TITLE (PROVISIONAL)	Burden of Mental Health Problems Among Pregnant and Post-Partum Women in Sub-Saharan Africa: Systematic Review and Meta-analysis Protocol
AUTHORS	Awini, Elizabeth; Agyepong, Irene; Owiredu, David; Gyimah, Leveana; Ashinyo, Mary; Yevo, Lucy; Aye, Sorre Grace Emmanuelle Victoire; Abbas, Shazra; Cronin de Chavez, Anna; Kane, Sumit; Mirzoev, Tolib; Danso-Appiah, Anthony

VERSION 1 – REVIEW

REVIEWER	Othieno, Caleb University of Nairobi, Psychiatry
REVIEW RETURNED	29-Jan-2023

GENERAL COMMENTS	Thanks for the comprehensive protocol. It is indicated that no ethical review is required, however I would recommend seeking an exemption from the relevant IRB.
---

REVIEWER	Owusu-Antwi, Ruth Komfo Anokye Teaching Hospital, Department of Psychiatry
REVIEW RETURNED	13-Feb-2023

GENERAL COMMENTS	The limitation was not adequately discussed. Apart from simply stating that this review is the first to attempt to rigorously summarize the burden of MMH problems in SSA, the exact exact limitations, like there were no previous data to be compared to, etc was not clearly stated.
---

REVIEWER	Catsaros, Stephanie Université Paris Cité
REVIEW RETURNED	22-Mar-2023

GENERAL COMMENTS	This is a very ambitious and crucial review to be done. However, I have several comments I would like to share: 1. The abstract does not clearly refer to the research outcomes that are being investigated. 2. Authors do not put any language limitations for the search string: is this feasible? 3. The search string will apply from “inception” to nowadays: if the search findings refer to a time period far revoked, for example an article published 30 years ago, how will this data relate to the actual settings of every country? It seems to me that If used, this data should be strictly and clearly time-framed, and contextualized (notably for exterior factors like a period of war for instance, during which PTSDs can be far more common).
---

	4. Authors refer to “predispositions” only for socio demographic characteristics and obstetrical history and characteristics. It seems that psychiatric predispositions are also a very important issue, especially when considering perinatal mental health burden. It would be interesting to clearly differentiate preexisting mental health issues from those triggered during or after pregnancy. This differentiation would most likely lead to different ways of implementing preventive care and interventions. 5. The protocol does not specify if only quantitative or quantitative and qualitative studies will be included. I wish the authors all the best for this very interesting work.
--	--

VERSION 1 – AUTHOR RESPONSE

Comments by Reviewer 1

It is indicated that no ethical review is required, however I would recommend seeking an exemption from the relevant IRB.

This systematic review is part of a larger study on maternal mental health which has received ethical clearance from the ethics review committee of the Ghana Health Service (GHS-ERC 012/03/20). We have added the statement below to the “Ethics, dissemination and protocol registration” and the main document, page 3.

Although no ethical clearance or exemption is needed for a systematic review, this review is part of a larger study on maternal mental health that involves primary data collection and which has received ethical clearance from the ethics review committee of the Ghana Health Service (GHS-ERC 012/03/20).

Comments by Reviewer 2

The limitation was not adequately discussed. Apart from simply stating that this review is the first to attempt to rigorously summarize the burden of MMH problems in SSA, the exact limitations, like there were no previous data to be compared to, etc was not clearly stated.

We thank Reviewer 2 for the comment. We have added a potential limitation of the study to the “Strengths and limitations” (main document, bullet 4).

A limitation of this systematic review is, there were no previous data to compare our systematic review to.

Comments by Reviewer 3

This is a very ambitious and crucial review to be done.

We thank Reviewer 3 for the kind comment.

However, I have several comments I would like to share:

The abstract does not clearly refer to the research outcomes that are being investigated.

We have added the primary research outcomes (prevalence and incidence) to the methods of the

abstract.

Authors do not put any language limitations for the search string: is this feasible?

On the question of whether this is feasible by not putting any language limitations for the search string, yes this is feasible. We have experts who can translate studies published in French or other languages into English. We did not place a language restriction because Sub-Saharan Africa has only two main languages, English and French (with Portuguese as a minor Language) and most articles are published in English, but also French. Placing a language restriction will lead to the exclusion of potentially relevant studies that may introduce a bias in the systematic review.

The search string will apply from “inception” to nowadays: if the search findings refer to a time period far revoked, for example an article published 30 years ago, how will this data relate to the actual settings of every country? It seems to me that If used, this data should be strictly and clearly time-framed, and contextualized (notably for exterior factors like a period of war for instance, during which PTSDs can be far more common).

We thank Reviewer 3 for this comment. We agree the research findings referring to a time period far remote may not reflect the current situation or context. The contexts of Sub-Saharan African countries vary substantially (e.g. independence, apartheid, wars) and to avoid unnecessary restrictions and missing potentially important data we decided to capture everything from inception.

Authors refer to “predispositions” only for socio demographic characteristics and obstetrical history and characteristics.

We thank Reviewer 3 for this comment. We have added another outcome to the secondary outcomes, please see below and in the revised manuscript (Secondary outcomes, bullet 3).

Psychiatric predisposition (pre-existing mental health issues triggered during or after pregnancy).

The protocol does not specify if only quantitative or quantitative and qualitative studies will be included.

This is quantitative systematic review. We have clarified this by adding “and Meta-analysis” to the title.